

# Continuous Light Absorption Photometer for Long-Term Studies

John A. Ogren[1,2], Jim Wendell[1], Elisabeth Andrews[2], and Patrick J. Sheridan[1]

[1]Global Monitoring Division, Earth System Research Laboratory, National Oceanic and Atmospheric Administration, Boulder, 80305, USA

[2]Cooperative Institute for Research in Environmental Sciences, University of Colorado, Boulder, 80303, USA

*Correspondence to*: John A. Ogren (john.a.ogren@noaa.gov)

**Abstract.** A new photometer for continuous measurements of aerosol light absorption coefficient, optimized for long-term studies of the climate-forcing properties of aerosols, is described. Measurements of the light attenuation coefficient are made at blue, green, and red wavelengths, with a detection limit of 0.02 Mm$^{-1}$ and a precision of 4% for hourly averages. The uncertainty

of the light absorption coefficient is primarily determined by the uncertainty of the correction scheme commonly used to convert the measured light attenuation to light absorption coefficient, and ranges from about 20% at sites with high loadings of strongly-absorbing aerosols up to 100% or more at sites with low loadings of weakly-absorbing aerosols. Much lower uncertainties (ca. 40%) for the latter case can be achieved with an advanced correction scheme.

## 1 Introduction

Reliable measurements of aerosol light absorption are crucial for quantifying the radiative forcing of climate. As a consequence, light absorption measurements are recommended for all stations in the Global Atmosphere Watch network, which is coordinated by the World Meteorological Organization. Aerosol light absorption is often dominated by soot-like particles produced by the incomplete combustion of carbonaceous fuels, commonly termed "black carbon". There are two basic approaches for determining the aerosol light absorption coefficient, *suspended-state* methods where the absorption is measured while the

particles are suspended in air, and *filter-based* methods where the particles are deposited on a filter for real-time optical analysis. Suspended-state methods are inherently more accurate because the physical state of the particles can be changed by the act of filtration, although both are subject to sampling artifacts associated with bringing particles from ambient conditions into a laboratory. Unfortunately, currently available instruments for suspended-state measurements are more expensive, less sensitive, and more difficult to operate than filter-based methods, making them better suited for use in intensive field campaigns or as

laboratory reference instruments. Filter-based instruments, because of their lower cost and simpler operation, have been the preferred choice for long-term measurements of aerosol light absorption in monitoring networks.

NOAA has used filter-based instruments for measuring aerosol light absorption coefficient at baseline observatories for over two decades. Both Aethalometers (Hansen et al., 1984) and Particle/Soot Absorption Photometers (PSAP; Bond et al., 1999) have been used. Both of these instruments, as well as the Multi-Angle Absorption Photometer (MAAP; Petzold and Schönlinner,

2004), lack one or more desirable features for long-term measurements of aerosol light absorption coefficient for studies of aerosol forcing of climate. For example, the MAAP only measures at single wavelength, the PSAP can require frequent (hourly to daily) filter changes, and the Aethalometer doesn't yet have a widely accepted correction scheme.

To address these issues, NOAA developed and built a filter-based instrument, the Continuous Light Absorption Photometer (CLAP), with the following design features:





- operation at three visible wavelengths (blue, green red), to allow calculation of the spectral dependence of key climate forcing parameters (aerosol single-scattering albedo and radiative forcing efficiency) when combined with a three-wavelength integrating nephelometer;

- high-sensitivity, for operation in relatively clean air;

- heated and temperature-stabilized, to minimize effects of changing room temperature and high ambient dewpoint temperatures;

- multiple filter spots, to enable unattended sampling for a week or more in rural and remote locations;

- internal flow paths optimized for low losses of particles smaller than 10 mm aerodynamic diameter;

- precisely-defined filter spot areas;

- optical configuration and filter media comparable to the PSAP, to allow use of the Bond et al. (1999) correction scheme for errors caused by filter loading and multiple scattering;

- low cost and small size.

Unlike the MAAP, the CLAP does require a co-located aerosol light scattering or extinction measurement to derive aerosol light absorption.

This paper describes the CLAP Model 10, so numbered because it was originally designed in 2010. As the original design evolved during development, several small adjustments were made to the prototype to provide the above features, including

- a two-part design, where the top half must be manually removed to change the filter. This approach was chosen to keep the mechanical design simple by eliminating the need for moving parts.

- elimination of o-rings to seal the filter. The initial design included an o-ring for each filter spot, but was rejected because the
o-ring caused the edge of the spot to be diffuse and variable.

- a redesigned voltage regulator to eliminate failures encountered in the field.

- use of a torque driver to secure the top and bottom assemblies after a filter change. The initial design used thumbscrews, but their uncontrolled torque sometimes resulted in incomplete sealing of the two assemblies.

The CLAP contains an embedded microprocessor to measure signals and control valves and heaters, but for simplicity the task of
calculating light absorption coefficient is delegated to an external computer. A simple user interface and menu system is provided for manual or computerized control of all CLAP functions.

## 2 Instrument Description

The CLAP belongs to the family of photometers that measure the transmission of light through a light-diffusing filter while particles are deposited on the filter. Other members of the family include the PSAP, MAAP, Aethalometer, and the Continuous
Soot Monitoring System (COSMOS; Kondo et al., 2009). Silicon photodiodes in the CLAP measure the intensity of diffuse light transmitted through the sample spots on the filter ($I_s$), and a second photodiode measures the intensity of light through an unsampled area of the filter ($I_r$). The logarithm of the ratio of the two intensities yields the attenuation,

$$ATN = -ln\left(\frac{I_s}{I_r}\right), \tag{1}$$

and the time rate of change of attenuation yields the attenuation coefficient ($\sigma_{atn}$, m$^{-1}$),

$$\sigma_{atn} = \frac{A}{Q}\frac{\left(ATN(t_2)-ATN(t_1)\right)}{(t_2-t_1)} = \frac{A}{Q}\frac{\Delta ATN}{\Delta t}, \tag{2}$$

where $ATN(t)$ is the filter attenuation at times $t_1$ and $t_2$ (in seconds), $Q$ (m$^3$ s$^{-1}$) is the sample flow rate through the filter, and $A$ (m$^2$) is the area of the exposed spot on the filter.





The aerosol light absorption coefficient $\sigma_{ap}$ is derived from the attenuation coefficient after corrections for changes in attenuation caused by light scattering from particles collected on the filter, multiple scattering and absorption of light within the filter medium, and reduction of the multiple scattering effect as the filter attenuation increases. These corrections for the PSAP were derived by Bond et al. (1999) and further elaborated by Ogren (2010):

$$\sigma_{ap} = 0.85\frac{f(\tau)\,\sigma_{atn}}{K_2} - \frac{K_1\sigma_{sp}}{K_2},$$  (3)

where $\sigma_{sp}$ is the aerosol light scattering coefficient adjusted to the wavelength of the absorption measurement. The transmittance correction term is defined as

$$f(\tau) = (1.0796\tau + 0.71)^{-1},$$  (4)

where $\tau = (I_s(t)/I_r(t))/(I_s(0)/I_r(0))$ is the normalized filter transmittance at time $t$ relative to transmittance at the start of sampling ($t=0$). The constants in Eq. (3) were derived by Bond et al. (1999) as $K_1$=0.02±0.02 and $K_2$=1.22±0.20, where the uncertainties are given for the 95% confidence level. The transmittance correction term, $f(\tau)$, and $K_2$ correct for the effects of filter loading and multiple scattering enhancement of absorption by particles within the filter matrix, while the $K_1$-term corrects for the change in attenuation caused by light scattering particles. The uncertainty of $f(\tau)$ was not directly reported by Bond et al. (1999), but their approach implicitly included that uncertainty, as well as the uncertainty in $\sigma_{sp}$, in the uncertainties of $K_1$ and $K_2$. An alternate correction scheme for the PSAP was reported by Virkkula et al. (2005).

The CLAP differs from the PSAP in that it utilizes solenoid valves to cycle through 8 sample filter spots and 2 reference (i.e., unsampled) filter spots, enabling the instrument to run at ideal conditions (filter transmittance, $\tau$, greater than 0.7) eight times as long as the single-spot PSAP. The CLAP was designed to use 47-mm diameter, glass-fiber filters (Pallflex type E70-2075W), identical to the original PSAP filters except for size. These filters are made of two fibrous layers, borosilicate glass fibers overlaying a cellulose fiber backing material (for strength and stability). The cellulose fiber layer may take up water under conditions of high humidity, which is one reason that the CLAP has an internal heater to lower the sample relative humidity inside the instrument.

A photograph and cutaway drawing of the CLAP optical and flow paths, as well as the internal configuration, are shown in Fig. 1; an annotated version of Fig. 1b is included in the Supplement (Fig. S1). The round "top hat" contains the white hemisphere and LED light source, with the sample inlet tube located at the center. A manifold in the upper plate directs the sample flow to one of eight sample spots on the filter below; miniature solenoids in the bottom part select the active sample spot. An external tube returns the filtered sample flow to the upper plate where it flows through one of the two reference spot to provide the measurement of $I_r$. The reference measurement alternates between the two reference spots because experience with the prototype instrument revealed that ATN drifts excessively during the first 10-20 minutes after a fresh sample spot is selected if the same reference spot is used. This drift is minimized if the reference spot is alternated, presumably because some time is needed for the reference spot to relax from the stretching caused by air flow through the filter. By switching the reference spot each time the sample spot is changed, both will stretch roughly the same when exposed to the flow regime. If there were only one reference spot, that spot would already be stretched, leading to large ATN drift until the sample spot reached a similar stretched state.

Diffuse sample illumination is provided by three sets of upward-facing light-emitting diodes (LED) in the upper half of the instrument. The LEDs illuminate the concave side of a white hemisphere, and the upward-facing detectors, in the lower half of the instrument, view this hemisphere through the sample deposit on the fiber filter. Because a parallel measurement of aerosol light scattering coefficient is required for calculation of the light absorption coefficient (Eq. 3), the wavelengths of the LEDs should match the wavelengths of the co-located scattering measurements as closely as possible. However, LEDs of sufficient





output intensity were not available at the TSI nephelometer wavelengths of 450, 550, and 700 nm, leading to the selection of LEDs with wavelengths of 468, 529, and 653 nm (see below).

A Texas Instruments MSP430F2618 microprocessor provides the minimum functionality for measuring signals (light intensity reaching the ten detectors, flow rate, case temperature, and sample temperature) and controlling the hardware (light source, case heater, and solenoid valves). The LEDs are cycled through four states (blue, green, red, dark) each second. Analog voltages from the detectors are digitized with 20-bit analog/digital converters (Texas Instruments DDC112); oversampling of the A/D converter increases the effective resolution to 22 bits. During development of the prototype CLAP, it became clear that noise levels of 1 Hz light intensities were unacceptably high, and that some smoothing was needed. Consequently, a digital low-pass filter was implemented in the microprocessor software; the default filter is a four-stage, single-pole design with an effective first-order time constant of 2.6 s.

The internal software detects when the pushbutton on the front panel is depressed and controls whether a red indicator lamp is lit or not; this lamp is off during normal sampling and on during a filter change. A blinking lamp indicates an error condition that must be corrected before continuing. A digital panel meter displays the sample flow rate, which is controlled by a needle valve. Sensor calibration and configuration parameters are stored in non-volatile memory, and are accessible through a menu-based user interface via the RS232C serial port. An external computer is required for data logging, instrument control (e.g., switching to the next filter spot), and calculation of the attenuation coefficient.

Four circuit boards hold all the electronic components, one square board in the upper half for the LEDs and associated circuitry, and three round boards in the lower half for the photodetectors, analog/digital converters, microprocessor, and associated circuitry. A Honeywell AWM43600V mass flowmeter is used to measure the sample flow rate. After each filter change, the upper half is secured to the lower half by tightening the four nuts to a torque of 2.5 N-m.

At NOAA federated network sites the CLAP is typically installed so that it draws its sample air through a modified TSI nephelometer (TSI, Inc., Model 3563) blower bypass block. A drawing of the modified blower block is included in the Supplement (Fig. S2). This set up allows for the CLAP zero reading to be measured each time a nephelometer background check with filtered air is performed (typically hourly at NOAA network sites). Vacuum for the CLAP is provided by an external pump.

## 3 Characterization

### 3.1 Particle sampling efficiency

The internal flow paths through the CLAP were chosen to minimize losses of particles in the size range of 0.01-10 μm. Internal flow velocities at the design volumetric flowrate of 1.0 lpm were 0.5-1 m s$^{-1}$, with flow Reynolds numbers of 230-300. A simplified model of the flow through the CLAP was used to estimate particle losses due to diffusion, impaction, and sedimentation. The combined particle sample efficiency shown in Fig. 2 indicates that particle losses are less than 10% for particles with aerodynamic diameters of 0.005-7 um, and less than 1% for 0.03-2.5 μm particles.

### 3.2 Wavelength response

The spectral intensity of the light source for each CLAP was measured with an Ocean Optics HR4000CG-UV-NIR spectrometer. An example of the normalized spectral intensity is shown in Fig. 3, along with the spectral response curve of the Hamamatsu S2386-18L detectors. Table 1 summarizes the measured output of the light sources. The spectral response of the detectors is sufficiently constant across the spectral output of the LEDs that the effective wavelengths of the three channels are within 1-2 nm of the values given in Table 1.



### 3.3 Spot area

Spot areas were determined by an automated digital analysis of photographs of exposed filters. The filters were placed on a grid with 5 mm line spacing, and photographs were taken with a high-resolution digital camera (12-16 megapixels). The lens was an Olympus 60 mm f2.8 Macro (120 mm equivalent). The analysis program determined the pixel locations of the grid intersection points and the approximate center of each spot, and then identified the outline of each spot based on the edge contrast. The area of each spot was calculated as the product of the number of pixels inside the outline and the size of each pixel. An example of the resulting analysis is shown in Fig. 4.

Each pixel in the 12 megapixel images was 23.8 μm square. Visual examination of the results of the automated image analysis revealed that the edge of the spots was identified within one pixel, i.e., the uncertainty of the measured spot radius was 23.8 μm. This uncertainty corresponds to an uncertainty of the spot area of 1.9%. Later images were taken with 16 megapixel camera, yielding a spot area uncertainty of 1.6%. For subsequent uncertainty calculations, we used a conservative estimate of the spot area uncertainty of 2%. The average area of 248 spots from a total of 31 individual instruments was 19.9 mm$^2$ with a coefficient of variation of 2.6%. The averaged measured area is 14% greater than the area of the 3/16-inch diameter holes that define the spots, suggesting that there may be a slight side leakage flow in the fiber filters. Another possible reason for the slightly larger spot size is that the holes were de-burred, yielding spot areas slightly larger than the machined hole. These possibilities reinforce the importance of measuring the spot areas rather than relying on internal dimensions.

### 3.5 Precision

Nine CLAPs of various ages (0.5-6 years) were operated in parallel in our laboratory for 15 days, sampling from a mixing chamber that was connected to outside air. The precision was calculated as the slope of the line relating the standard deviations of each set of nine values of attenuation coefficient to the means of the nine values (Fig. 5). The slopes for both 1-minute and 1-hour averages of the attenuation coefficient measured by these nine instruments were around 4% (Table 2). An alternative approach for calculating precision, using the means of the coefficients of variation for each minute or hour, yielded similar results.

The nine CLAP mass flowmeters were calibrated prior to the experiment with a BIOS Definer 220M flow calibrator. The coefficient of variation of repeated flow measurements with this calibrator in our laboratory was measured to be 0.2%, which indicates that the flow calibration is not a major contributor to the overall CLAP precision. The calibrator manufacturer's reported accuracy of 1% is irrelevant for these precision tests because the same flow calibrator was used for all instruments. However, the flow calibrator uncertainty should be included when calculating the overall uncertainty of CLAP measurements.

### 3.6 Noise

Noise characteristics of each CLAP were measured as part of the manufacturing process, as well as on occasions thereafter when instruments were returned for servicing. In a typical noise test, CLAP sample intensities and flowrates were recorded at 1 Hz for about eight hours on each spot while the instrument sampled filtered room air. The resulting time series for each spot was randomly sampled 500 times for each sample interval $\Delta t$, and the attenuation (Eq. 1) and attenuation coefficient (Eq. 2) were calculated for each 1-s time step. The average attenuation coefficient was calculated in two ways for each randomly-chosen sample:

"arithmetic" – the 1-Hz attenuation coefficients were averaged over the sample interval $\Delta t$;





"difference" – the 1-Hz attenuation at the start and end times of the sample interval $\Delta t$ was used to calculate the average attenuation coefficient from Eq. (2).

The standard deviation of the 500 random samples is interpreted as the measurement noise associated with the averaging time $\Delta t$. Results from this analysis are shown as the thick black line in Fig. 6 for the arithmetic averages; the corresponding results for the

difference-based averages were indistinguishable from the arithmetic averages and are not shown. Furthermore, the results for the three wavelengths were very similar and so the figure shows the combined statistics for all wavelengths. The results represent a total of 3778 hours of measurements using 28 different instruments. The statistics from this analysis are unreliable for averaging times longer than about 3 hours because the noise runs were generally around eight hours long. The error bars indicate the ±1 standard deviation range of the results from all the different spots, instruments, and wavelengths for each averaging time;

the lower bars for the rightmost points are not shown because the standard deviations were greater than the means for averaging times greater than 10,000 s (3 h)

A second analysis using five instruments, operated for about five weeks on five separate spots (one week/spot) was used to evaluate the instrument noise for longer averaging times. Input data for this experiment consisted of 1-minute, arithmetically-averaged attenuation coefficients. The results are plotted in Fig. 6 as the thin black line with open symbols. The overlap between

the two experiments for averaging times of 60-10,000 seconds indicates the consistency of the two approaches.

The slope of the curves in Fig. 6 is about -1 for averaging times of 10-100 s (red line), and about -0.5 for averaging times of 2-1440 minutes (blue line). For general use, the noise level of the CLAP attenuation coefficient can be approximated as 0.10 Mm$^{-1}$ * $(\Delta t / 100\ s)^n$ , with n = -1 for 5 s < $\Delta t$ < 100 s and n = -0.5 for 100 s < $\Delta t$ <= 24 h. Overall statistics for 1-minute averages yield a noise level of 0.19 Mm$^{-1}$ with a coefficient of variation of 38% for the 28 instruments tested (cf. 0.17 Mm$^{-1}$ using the algorithm

above).

Müller et al. (2011) reported noise levels of absorption coefficients from six, three-wavelength PSAPs of 0.06-0.07 Mm$^{-1}$ for 1-min averages for all three wavelengths. The corresponding noise levels, in terms of the attenuation coefficient, are 0.13-0.15 Mm$^{-1}$, i.e., slightly lower than the CLAP value of 0.19 Mm$^{-1}$. However, Müller et al. (2011) used an averaging method that was based on 1-min averages of the sample intensities, i.e., the 1-min averages of the PSAP attenuation coefficient used data from

two consecutive minutes. As a consequence, it is appropriate to compare the noise from 2-min average attenuation coefficients for the CLAP (0.11 Mm$^{-1}$) to the 1-min averages from the PSAP noise tests (ca. 0.14 Mm$^{-1}$).

Springston and Sedlacek (2007, hereafter SS07) presented a comprehensive analysis of the PSAP noise characteristics. Many of their results are specific to the peculiarities of the internal data processing and serial output of the PSAP and are not applicable to the CLAP. In fact, one of the design goals for the CLAP was to implement straightforward internal data processing and high-

resolution serial data reports in order to avoid the limitations of PSAP serial data reports. However, the results in Fig. 6 are comparable to the results for Case II reported in Fig. 4 of SS07, which show a noise level for 1-min averages of the single PSAP tested of 0.05 Mm$^{-1}$. That value of 0.05 Mm$^{-1}$ needs to be multiplied by the transmittance correction factor $f(\tau)$ to convert it to the noise of the attenuation coefficient. Assuming that SS07 measured the noise on blank filters ($\tau$=1), their results correspond to a noise level of the attenuation coefficient of 0.09 Mm$^{-1}$, somewhat lower than the PSAP noise levels reported by Müller et al.

(2011). The slope of the regression line for the SS07 Case II is -1 for averaging times of 4-100 s, indistinguishable from the results for the CLAP shown in Fig. 6.

### 3.7 Measurement uncertainty

The uncertainty of the measured attenuation coefficient (Eq. 2) is given by





$$\frac{\delta\sigma_{atn}}{\sigma_{atn}} = \sqrt{\left(\frac{\delta\Delta ATN}{\Delta ATN}\right)^2_{noise} + \left(\frac{\delta\sigma_{atn}}{\sigma_{atn}}\right)^2_{precision} + \left(\frac{\delta Q}{Q}\right)^2}, \quad (5)$$

where $\delta X$ denotes the uncertainty of $X$. Eq. (5) assumes that the uncertainty of the measurement interval $\Delta t$ is negligible. The 2% uncertainty of the spot area is implicitly included in the 4% precision of the CLAP measurements, and should not be double-counted. As a result, the uncertainty of the measured attenuation coefficient, including the 4% precision and the 1% uncertainty

of the flow calibration, is calculated as

$$\frac{\delta\sigma_{atn}}{\sigma_{atn}} = \sqrt{\left(\frac{\delta\Delta ATN}{\Delta ATN}\right)^2_{noise} + (0.01)^2 + (0.04)^2} = \sqrt{\left(\frac{\delta\sigma_{atn}}{\sigma_{atn}}\right)^2_{noise} + (0.041)^2} \quad (6)$$

The noise term in Eq. (6) is calculated from the noise measurements described in Sect. 3.5, as shown in Fig. 6. Application of Eq. (6) for different attenuation coefficients and averaging times yields the relative uncertainty in the attenuation coefficient, expressed at the 95% confidence level, as a function of averaging time and attenuation coefficient (Fig. 7). For hourly averages

and attenuation coefficients larger than 2 Mm$^{-1}$, the uncertainty of attenuation coefficients measured by the CLAP is 8%, determined entirely by the flow calibrator accuracy and the CLAP precision.

The uncertainty of the absorption coefficient (Eq. 3) can be written as

$$\frac{\delta\sigma_{ap}}{\sigma_{ap}} = \frac{1}{K_2}\sqrt{(K_2 + aK_1)^2\left[\left(\frac{\delta\sigma_{atn}}{\sigma_{atn}}\right)^2_{noise} + (0.41)^2\right] + (a\delta K_1)^2 + (\delta K_2)^2}, \quad (7)$$

where $a = \varpi_0/(1 - \varpi_0)$ and $\varpi_0 = \sigma_{sp}/(\sigma_{sp} + \sigma_{ap})$ is the single-scattering albedo. The quantity in square brackets in Eq. 7 is

the uncertainty of the attenuation coefficient (Eq. 6), and the uncertainties of the parameters of the Bond et al. (1999) correction are $\delta K_1$=0.01 and $\delta K_2$=0.1 (at the 1-standard deviation confidence level, for consistency with Eq. 6). Fig. 8 shows the results of evaluating Eq. 6 for several different averaging times and single-scattering albedos. Fig. 8 demonstrates that the uncertainties associated with averaging time and attenuation coefficient are negligible, compared to uncertainties associated with the Bond et al. (1999) correction, for averaging times of five minutes or more and attenuation coefficients of 1 Mm$^{-1}$ or more. The results in

Fig. 8 are consistent with the results derived for the PSAP by Müller et al. (2014, Fig 14b).

Equation (7) shows that values of both the single-scattering albedo and attenuation coefficient are needed to derive the uncertainty of the absorption coefficient, which precludes giving a single value for the uncertainty of the absorption coefficient for ambient aerosol measurements. However, NOAA and collaborators have operated CLAPs at a variety of sites with a wide range of single-scattering albedos and attenuation coefficients, which allows calculation of the frequency distribution of the

uncertainty of the absorption coefficients measured at those sites. Fig. 9 illustrates the measurement uncertainty of the absorption coefficient at 95% confidence calculated using Eq. 7 for 30-minute averages, along with the inter-quartile range of single-scattering albedo and attenuation coefficient for eight sites. An averaging time of 30-minutes was chosen because these sites (except SUM) employ switched impactors to measure both PM10 and PM1 size ranges over the course of an hour, and so the hourly averages from these stations are based on about 30 minutes of data for each size cut. The station-specific results in Fig. 9

are combined values for all three wavelengths of the CLAP and for both PM10 and PM1 size ranges, for the four-year period 2013-2016. The lowest uncertainties are seen for stations with greater absorption and lower single-scattering albedo (e.g., ARN, GSN) and the highest uncertainties are seen when absorption is low and single-scattering albedo is high (e.g., BRW). The overall median measurement uncertainty for these eight stations is 30%, and 75% of the time the uncertainty is below 49%.

The uncertainties shown in Fig. 9 are primarily determined by the uncertainty of the $K_1$ and $K_2$ parameters of the Bond et al.

(1999) correction scheme. The constrained two-stream (CTS) correction scheme reported by Müller et al. (2014) yields much




lower uncertainties for weakly-absorbing aerosols, with uncertainties reduced to about 30% for a single-scattering albedo of 0.98.

**3.7 Comparison with PSAP**

CLAPs were deployed at NOAA and some collaborating partner stations beginning in 2010. A minimum of one year of parallel
operation with the existing PSAPs was required before the PSAPs were retired from service. The resulting data set for a statistical comparison of CLAP and PSAP measurements is comprised of 27 station-years of data from 17 stations. An example of the comparison for one station is shown in Fig. 10. The PSAP and CLAP filter changes were not synchronized for these comparisons, resulting in different transmittances reported by the two instruments. The transmittance correction factor $f(\tau)$ (Eq. 4) was applied to the attenuation coefficients calculated with Eq. (2) to put the two measurements on a comparable basis. The
regression analysis was done with a principal components technique that minimizes the orthogonal distance to the regression line so that uncertainties in both the x- and y-variables are considered. The final analysis excluded points that are more than three standard deviations from the regression line to reduce the sensitivity to outliers. Intercepts of the regression lines were generally small (mean of 0.1 Mm$^{-1}$) and so were neglected for the following analysis.

The analysis exemplified in Fig. 10 was repeated for all stations and wavelengths. The average slopes of the regression lines
were 0.97, 0.96, and 0.90 for the blue, green, and red wavelength channels, respectively; the overall mean slope was 0.94 with a standard deviation of 0.08. The regression slopes are plotted in Fig. 11 as a function of the mean attenuation coefficient.

The spectral outputs of the CLAP and PSAP light sources differ slightly (cf. Table 1 here and Table 6 in Müller et al., 2011). Assuming that the absorption coefficient is inversely dependent on wavelength, which is characteristic of black carbon, these slight differences would cause the absorption reported by the CLAP to be +1.1%, +1.7%, and -0.5% greater than the PSAP
results for the blue, green, and red measurement wavelengths, respectively. No correction for these slight differences as made in the CLAP-PSAP comparison reported here.

A comprehensive evaluation of the precision and noise level of PSAP measurements is not available, although the limited evaluations that have been published (Bond et al., 1999; Müller et al., 2011) suggest that the uncertainty of attenuation coefficients measured with the PSAP is probably similar to, and perhaps somewhat greater, than that derived here for the CLAP.
Assuming that the two measurements have a comparable uncertainty of 10%, disregarding the contribution of measurement noise, then the resulting uncertainty of the ratio of the two measurements is 14% (95% confidence bound). The average ratio of 0.94 derived here is well within the combined uncertainty of the two instruments.

**4 Operation with alternative filter**

The Pallflex E70-2075W filters are no longer commercially available so a replacement filter (model 371M, Azumi Filter Paper
Co., Japan) has been investigated. These filters were evaluated by Irwin et al. (2015) for use in the COSMOS. Irwin et al. (2015) developed an alternative transmittance correction $f(\tau)$ for COSMOS and compared results from 10-days of parallel operation of two COSMOS units in Tokyo equipped with Azumi and Pallflex filters. They reported that black carbon concentrations measured by COSMOS with Azumi filters were 6-8% higher than the values measured with Pallflex filters, depending on the transmittance correction used. The 6-8% difference in black carbon concentrations corresponds directly to a 6-8% difference in
transmittance-corrected attenuation coefficients.

The Azumi 371M filters are a spunbonded nonwoven fabric consisting of 41% glass fibers and 59% polyethylene terephthalate (PET) polymerized polyester fibers; less than 1% of polymerized fluorine is also present. The filter thickness is 0.31 mm, but the



fiber diameter is proprietary to the manufacturer. As a result of the manufacturing process, the polyester and glass fibers are randomly mixed, i.e., there is no "top" or "bottom" side. Without the cellulose backing, the Azumi filter may be less prone than the Pallflex filter to respond to changes in relative humidity.

Two different experiments were performed to compare the CLAP response to Azumi filters, relative to PSAP filters. These experiments do not represent a comprehensive evaluation of the CLAP response to Azumi filters, but rather a study of the feasibility of using Azumi filters in the CLAP; a more comprehensive evaluation will be the subject of a future paper. The first experiment sampled ambient air in Boulder, Colorado, while the second sampled nebulized and dried Regal Black particles (REGAL R400 pigment black, Cabot Corp., USA) in the laboratory. In both experiments, one CLAP used the Pallflex E70-2075W filter and the second CLAP used the Azumi 371M filter. The raw CLAP data were corrected in real-time using the measured spot areas and flow calibrations, and 1-minute averages were recorded. The slope of the regression line, forced through the origin, of attenuation coefficients measured on Azumi vs. Pallflex filters was calculated using three different transmittance corrections: no correction, $f(\tau)$ from the Bond et al. (1999) correction (Eq. 4), and the alternative corrections derived for COSMOS by Irwin et al. (2015). Separate equations were used for Pallflex and Azumi filters for the latter case (Equations 10 and 11, respectively, from Irwin et al., 2015). Linear regressions were performed on trimmed data, where only the central 95% of the data were considered. The linear regression line was forced through the origin, because the unforced regression lines all had intercepts very close to zero. Results obtained from the two filters were highly correlated, with coefficients of determination of 0.98 or greater.

The ambient measurements used in this comparison were made from Dec. 2, 2015 through Feb. 19, 2016. Approximately 100,000 one-minute average values were available for the comparison. The laboratory measurements on Regal Black were of much shorter duration, consisting of about 300 minutes of data from three runs when the transmittance on the Pallflex filters decreased from 1 to about 0.5.

The wavelength dependence of the regression slope of attenuation coefficient measured with Azumi vs. that measured with Pallflex was minimal, with the blue wavelength averaging 0.7% higher and the red wavelength averaging 1.4% lower than the slope for the green wavelength, for both the black and ambient aerosol experiments. For simplicity, and for comparison with Irwin et al. (2015), only results for the green wavelength are presented here.

Light attenuation coefficients measured on the Azumi filters were consistently about 25% greater than those measured on the Pallflex filters, regardless of the scheme used to correct for the dependence on instrument response on filter loading (Table 4). This difference is likely due to different physical and optical properties of the filters, e.g., pore size, fiber diameter, particle penetration depth, and multiple-scattering characteristics. The results from Boulder are consistent with Irwin et al.'s results from Tokyo in showing that the different correction schemes do not create a large difference in the ratio of light absorption coefficients (or EBC mass concentration). However, the Azumi/Pallflex light attenuation ratios measured with CLAPs in Boulder (ca. 1.25) are substantially greater than the ratios measured with COSMOS in Tokyo (ca. 1.05). Some possible hypotheses to explain the different results from Boulder compared to Tokyo are:

- the light absorbing particles in Boulder are different from that in Tokyo;

- sampling issues, such as different particle losses downstream of each instrument inlet;

- the powerful heater in COSMOS alters the particles in a way that changes the filter response;

- the optics in the CLAP and COSMOS are different.

Further experiments are needed to test these hypotheses.





**5 Conclusions**

The CLAP has proven to be a reliable instrument for measuring aerosol light absorption coefficient under a wide variety of sampling conditions, including from research aircraft. The two-part design, with the associated need to separate the two halves for each filter change, has been mostly problem-free. Likewise, the need for a torque driver to tighten the two parts together has

not been an operational hurdle. With its small internal heaters, sensitivity to sample temperature and relative humidity is much reduced compared to the PSAP. Filter changes are typically needed once per month at remote sites, once or twice per week at rural continental sites, and daily at very polluted sites.

As of mid-2017, 22 CLAPs are deployed at long-term monitoring sites operated by NOAA and its partners. A commercial version of the CLAP, called the Tri-color Absorption Photometer (TAP, Brechtel Manufacturing, Inc., Hayward, CA, USA), is

now available, and production of CLAPs at NOAA is expected to wind down as a result. Evaluation of the TAP response to laboratory and urban aerosols will be the subject of a future paper.

The Pallflex filter originally used in the CLAP and PSAP is no longer available. Limited tests with a replacement filter (Azumi 371M) show a high correlation between the two filter types, with the Azumi filter yielding results that are about 25% higher using the Bond et al. (1999) correction scheme. Further research is needed to assess whether a different correction scheme will

be needed for the Azumi filter.

**Code Availability**

Technical information on the CLAP, including construction drawings, schematics, printed circuit board layouts, and source code, are available upon request from Jim.Wendell@noaa.gov.

**Appendix A: Specifications**

| | |
|---|---|
| Wavelengths (centroid (FWHM)), nm | 467 (26), 529 (40), 653 (20) |
| Filter media | Pallflex type E70-2075W or Azumi type 371M, 47 mm diameter |
| Flowrate (volumetric), lpm | 1.0 |
| Noise (standard deviation of 60-sec averages of attenuation coefficient on filtered air), Mm$^{-1}$ | ~0.2, all wavelengths |
| Number of sample spots | 8 |
| Number of reference spots | 2 |
| Inlet connection | ¼ inch outer diameter tube |
| Outlet connection | Hose barb for 1/8 inch inner diameter tube |
| Dimensions (L x W x H), cm | 10 x 10 x 16, excluding back panel connectors |
| Weight, kg | 1.6 |
| Power consumption (heaters on) | 36 W @ 120-240 VAC, using supplied adapter |
| | 1.5 A @ 24 VDC |
| Power adapter | 11 x 5 x 2 cm, weight 0.15 kg |
| Mounting holes | Four holes in a square pattern centered on bottom, spaced 2.5" (6.35 cm) apart, tapped for 10-32 machine screws |
| Torque setting for tightening top and bottom | 2.5 |



| | |
|---|---|
| sections, N-m | |
| Serial communications | RS232, 57600 baud, no parity, 8 data bits, 1 stop bit. Data rate in unpolled operation is one record of 460 bytes sent each second (user-controllable interval) |

## Author contribution

J. Ogren designed the experiments and J. Ogren, P. Sheridan and E. Andrews carried them out. J. Wendell designed and built the instruments, and wrote the operating software. J. Ogren prepared the manuscript with contributions from all co-authors.

## Competing interests

The authors declare that they have no conflict of interest.

## Disclaimer

The scientific results and conclusions, as well as any views or opinions expressed herein, are those of the author(s) and do not necessarily reflect the views of NOAA or the Department of Commerce. Mention of a commercial company or product does not constitute an endorsement by the NOAA Global Monitoring Division. Use of information from this publication concerning

proprietary products or the test of such products for publicity or advertising purposes is not authorized.

## Acknowledgements

The authors are grateful to Wilfred von Dauster for taking the photograph for spot size determination, Derek Hageman for writing the data acquisition software and assisting with data analysis, Anne Jefferson for comments on the draft manuscript, Katy Sun for assistance in preparing figures, Mar Sorribas for providing data from ARN, Sang-Woo Kim for providing data from

GSN, and Neng-Huei Lin and the Taiwan EPA for providing data from LLN. Funding for this effort was provided by the NOAA Climate Program Office.

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



Table 1: Spectral output characteristics of light source. Values in parentheses are the coefficients of variation of measurements on 31 light sources. FWHM denotes the width of the curves at half the peak intensity.

|  | Wavelength at peak output (nm) | Wavelength at centroid (nm) | FWHM (nm) |
|---|---|---|---|
| Blue | 462.0 (0.3%) | 467.6 (0.5%) | 27.7 |
| Green | 522.3 (0.1%) | 528.7 (0.2%) | 39.8 |
| Red | 653.4 (0.1%) | 653.0 (0.1%) | 20.0 |

5    Table 2: Precision of CLAP attenuation coefficients for 1-minute and 1-hour averaging times

|  | 1 minute | 1 hour |
|---|---|---|
| Blue | 4.3% | 3.9% |
| Green | 4.3% | 3.9% |
| Red | 4.8% | 4.1% |

Table 3: Ratio of light absorption coefficient measured with Azumi filter to the value measured with Pallflex filter.

| Transmittance Correction | Ambient | Black |
|---|---|---|
| no correction | 1.25 | 1.21 |
| Bond et al. (1999) | 1.30 | 1.25 |
| Irwin et al. (2015) | 1.26 | 1.21 |



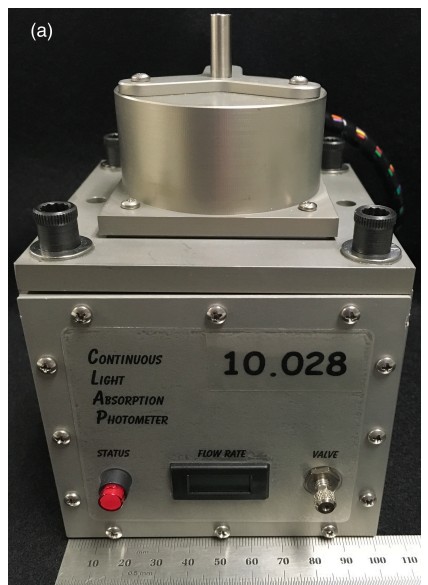
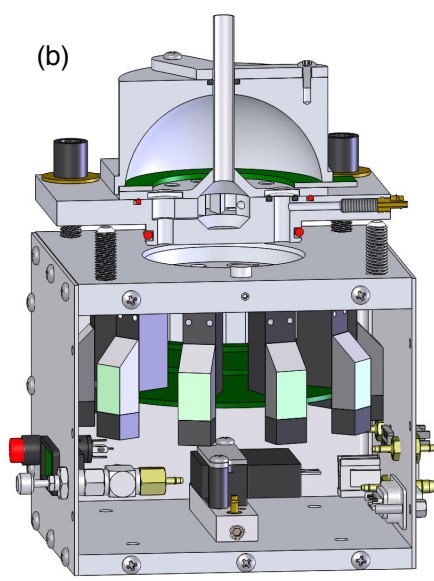

**Figure 1: (a) Photograph of CLAP, and (b) cross-sectional view of flow and optical paths in upper section and three-dimensional view of CLAP internal configuration in lower section. Scale on ruler in left image is in millimeters. An annotated version of (b) is included in the Supplement (Fig. S1).**

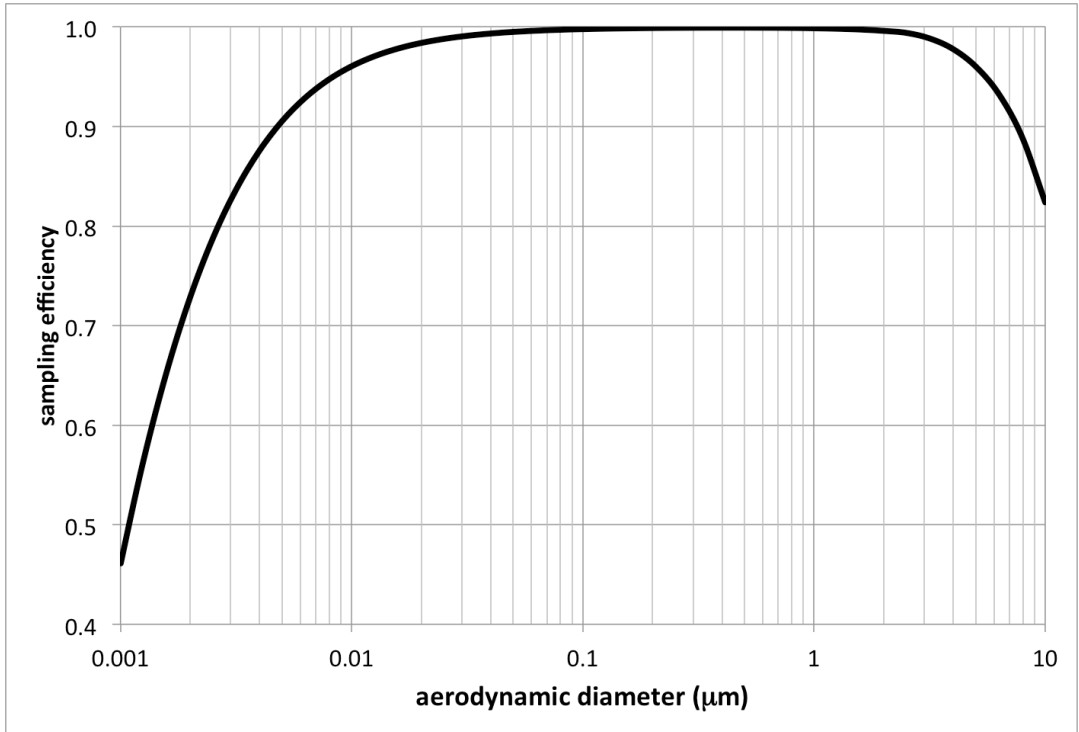

**Figure 2: Calculated sample efficiency of particles reaching the CLAP filter**




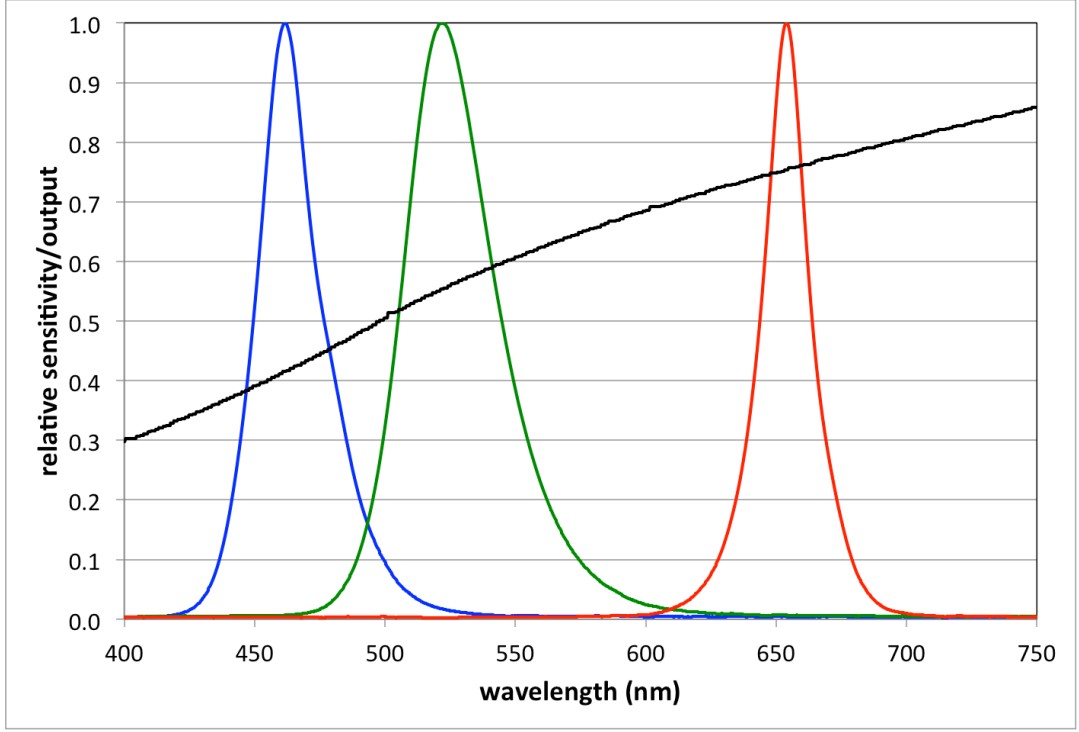

**Figure 3:** Normalized spectral output of light source for blue, green, and red channels, and spectral sensitivity of detector (black curve).





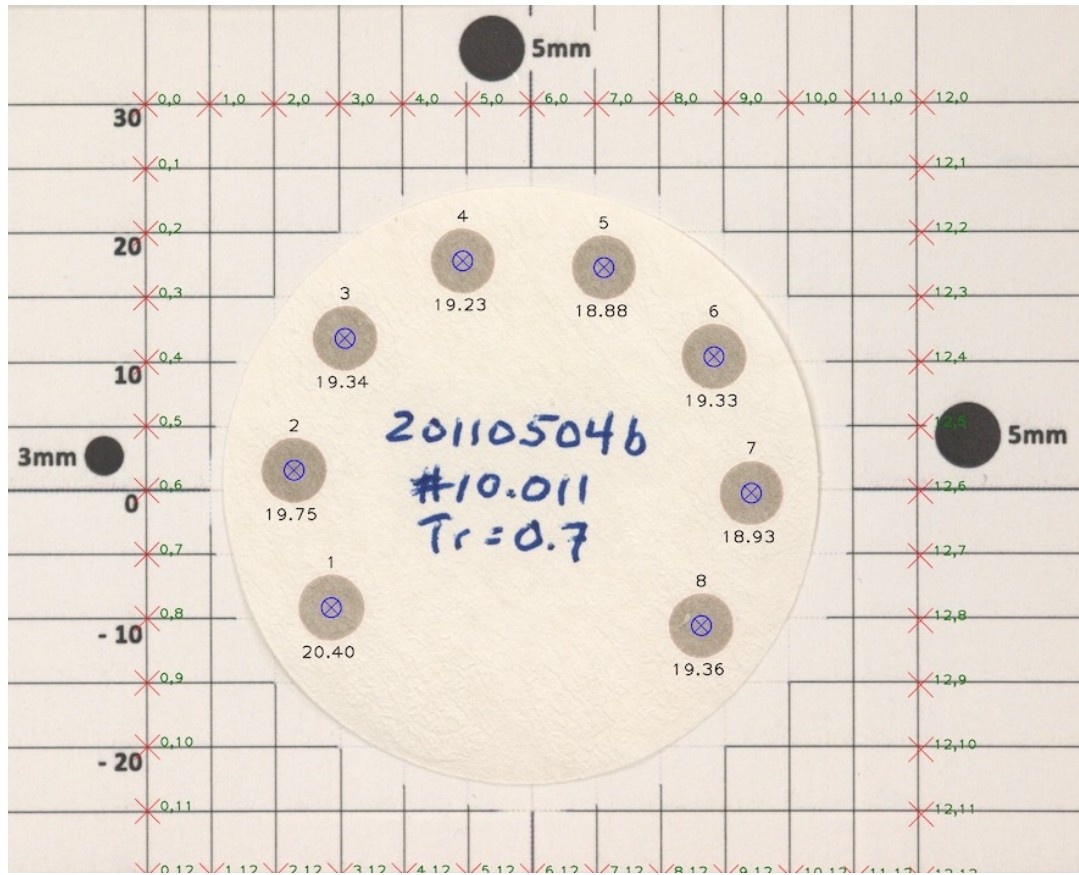

**Figure 4: Example of photographic analysis of spot area. Numbers 1-8 above each spot indicate spot number, while the numbers below each spot indicate spot area in mm$^2$.**




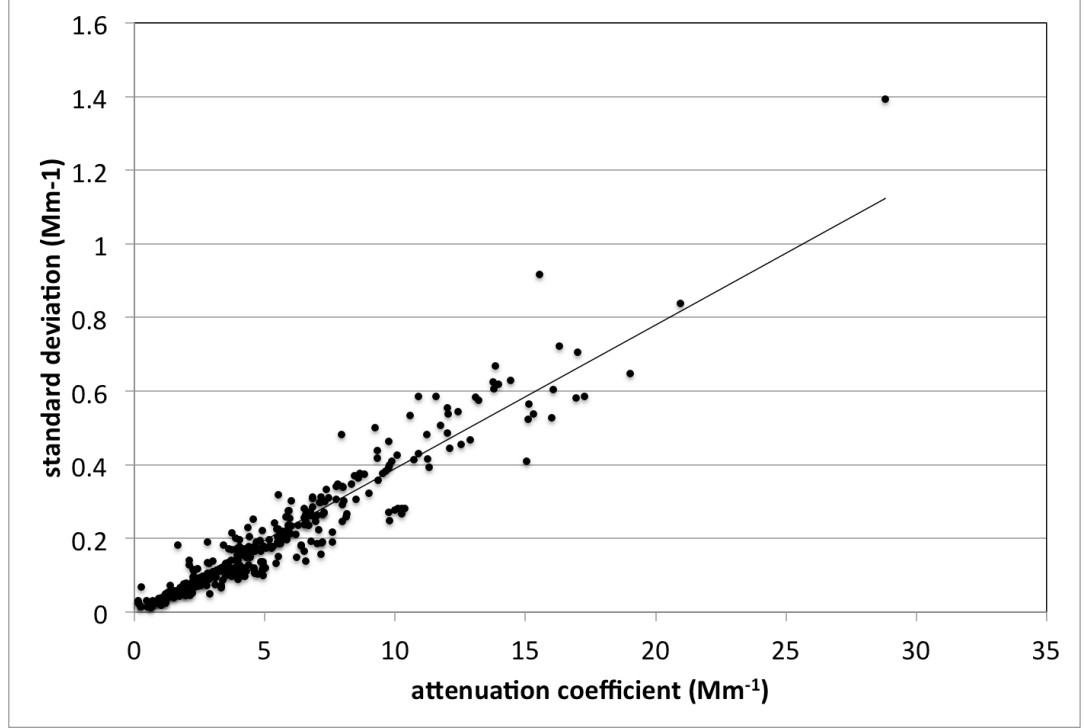

**Figure 5: Standard deviation of attenuation coefficient measured by nine CLAPs (1-hr averages, 529 nm wavelength). Line indicates slope of regression, forced through the origin.**



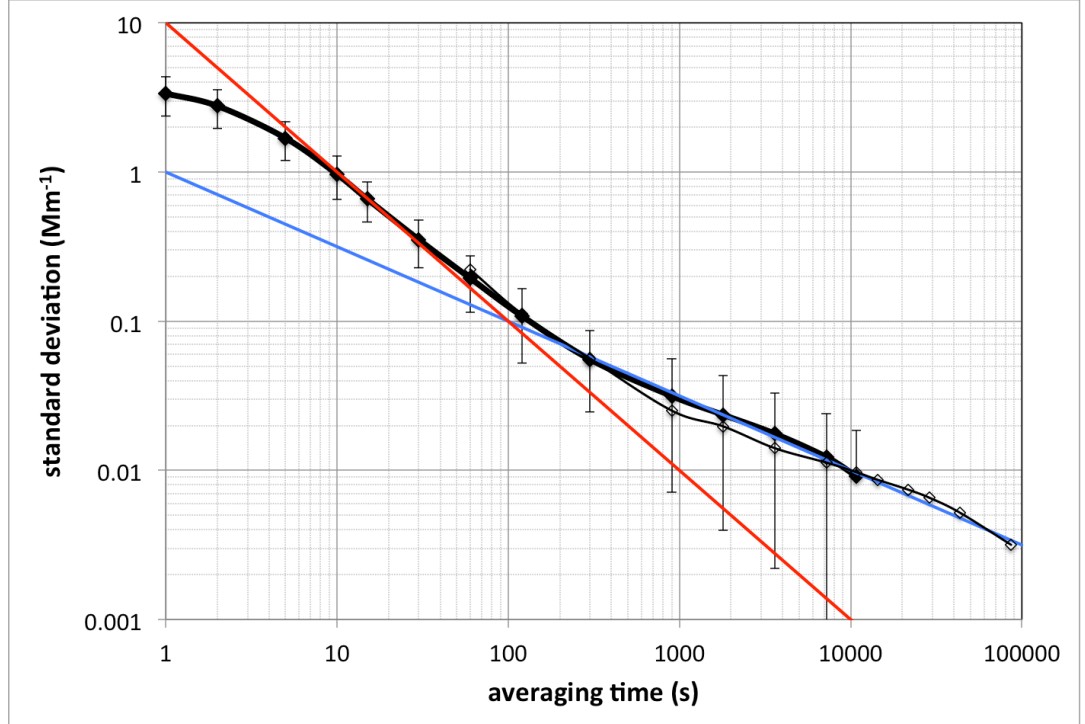

**Figure 6: Standard deviation of attenuation coefficient measured on filtered air, as a function of averaging time. Thick black line with solid symbols represents the measurement data based on 8 h of measurements/spot; thin black line with open symbols shows the measurement data when each spot was sampled for 1 week; red line represents approximate slope for 5-100 s averaging time; blue line represents approximate slope for 2 min to 1 day averaging times.**





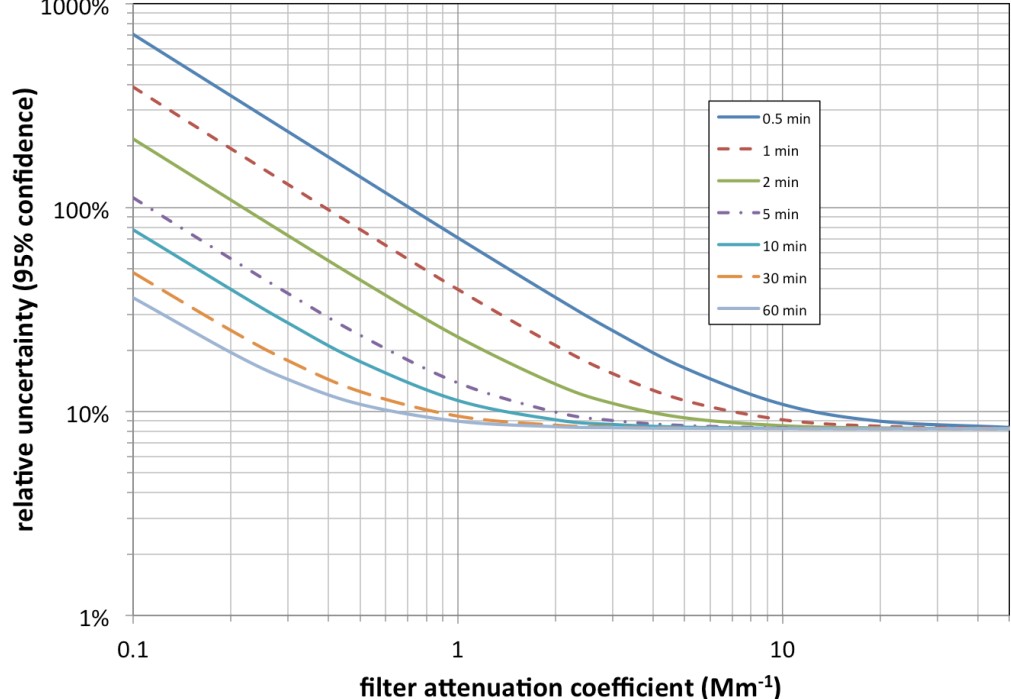

**Figure 7: Uncertainty of CLAP measurements of attenuation coefficient as a function of averaging time and attenuation coefficient, expressed as the 95% confidence level.**





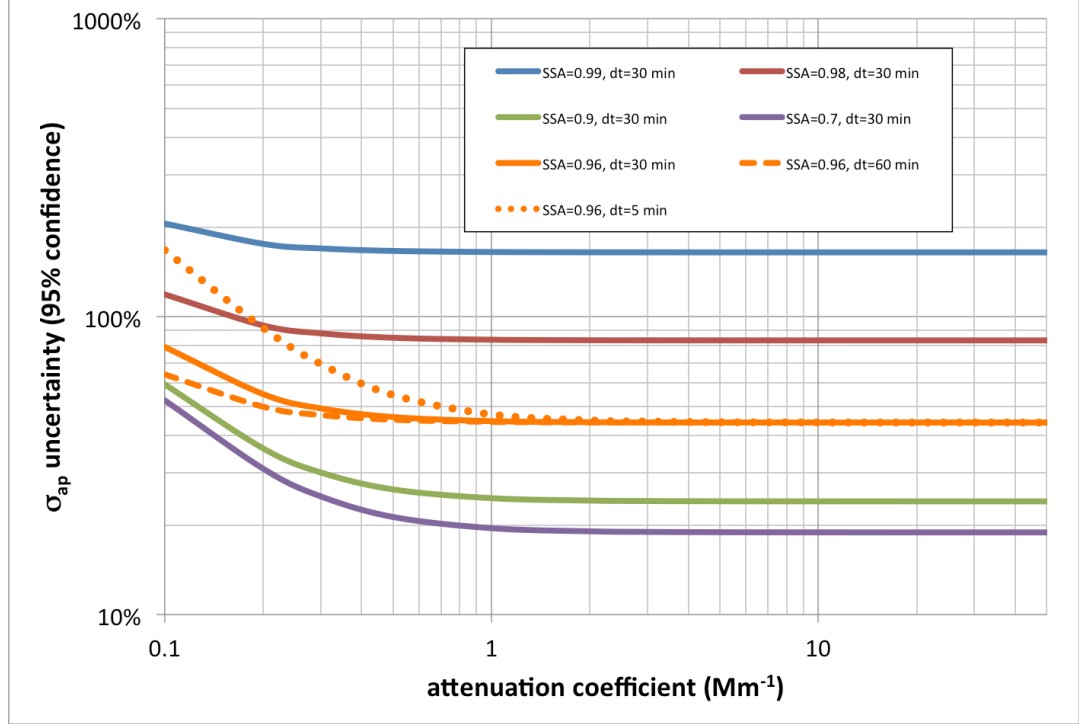

**Figure 8: Uncertainty of CLAP measurements of absorption coefficient (95% confidence level) as a function of attenuation coefficient, for various values of single-scattering albedo (SSA) and averaging time (dt). Solid lines are all for 30 min averaging time, dashed lines are for other averaging times as noted in the legend**





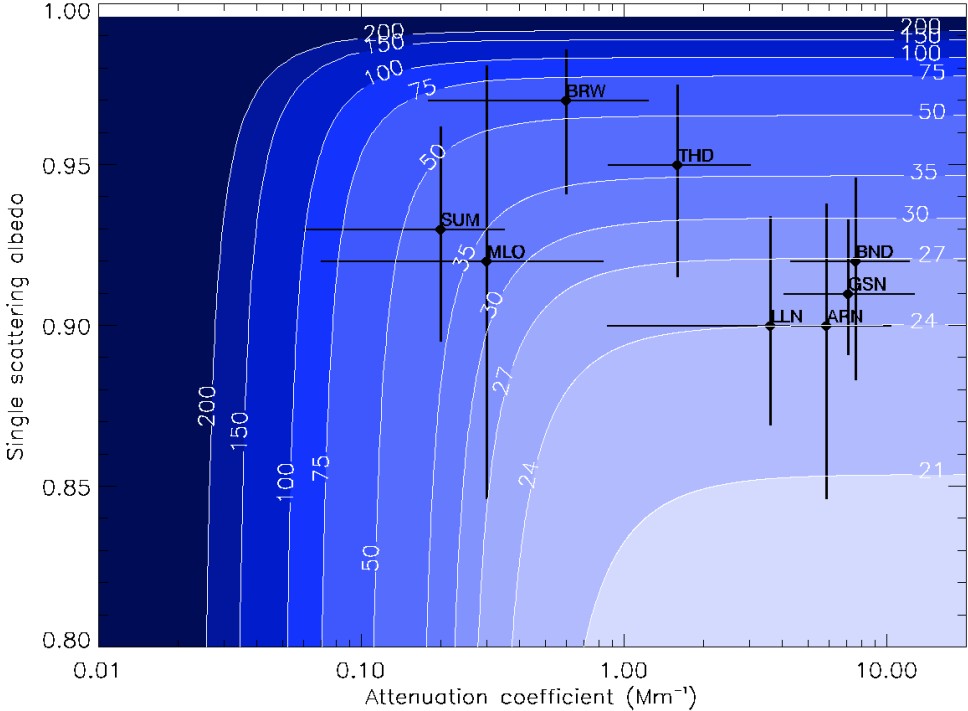

**Figure 9: Percent uncertainty of 30-min average light absorption coefficient as a function of single-scattering albedo and attenuation coefficient. The median uncertainty and inter-quartile range are shown for values measured for 2013-2016 at eight sites (ARN, El Arenosillo, Spain; BND, Bondville, USA; BRW, Barrow, USA; GSN, Gosan, South Korea; LLN, Mt. Lulin, Taiwan; MLO, Mauna**
5  **Loa, Hawaii; SUM, Summit, Greenland; THD, Trinidad Head, USA). Uncertainties are given at the 95-percent confidence level.**




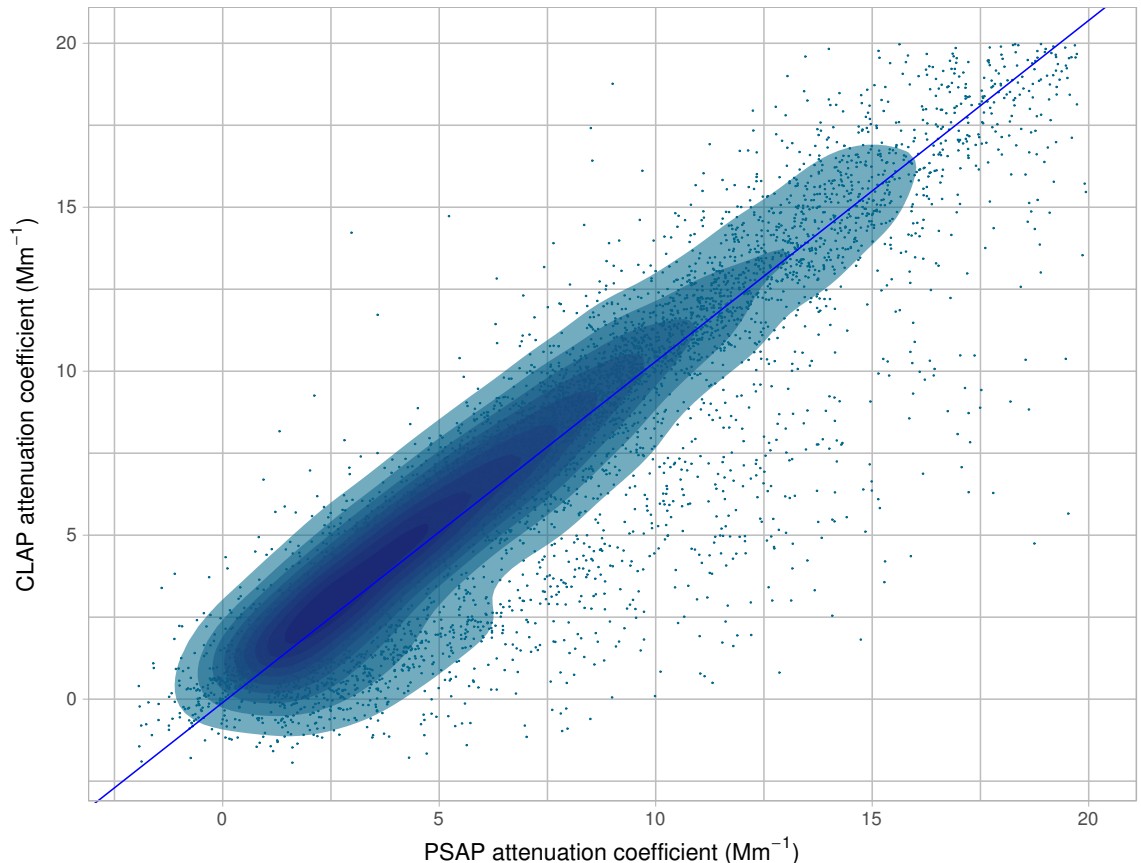

**Figure 10: Comparison of attenuation coefficients measured with CLAP and PSAP. Results represent 8442 30-minute averages for Bondville, IL, USA (PM10 size range, blue wavelength channel). The shaded areas contain 90% of the observations and the color shading distinguishes the deciles of the two-dimensional probability distribution function. The slope of the regression line is 1.04 with an intercept of -0.10; the regression line accounts for 98% of the variance in the measured data.**





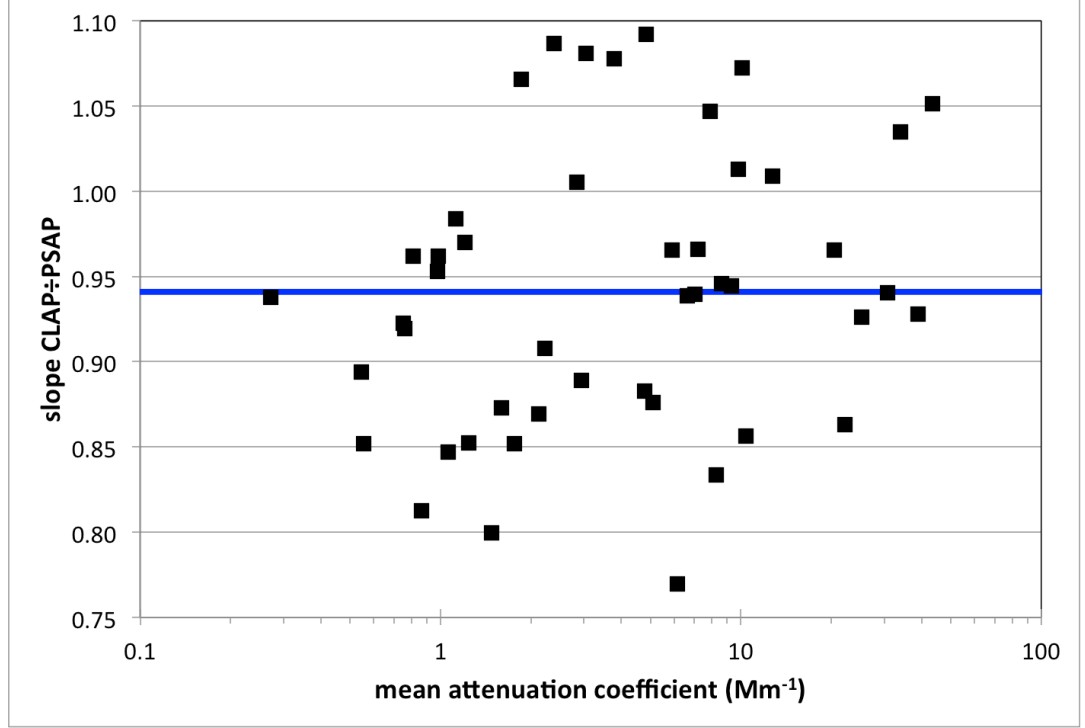

Figure 11: Slope of orthogonal regression line relating attenuation coefficients measured by CLAP vs. PSAP as a function of mean attenuation coefficient for 17 stations. The blue line indicates the mean slope of 0.941. Data are given for all measured wavelengths.