# Peer review of "Continuous Light Absorption Photometer for Long-Term Studies"

_Atmospheric Measurement Techniques, 2017_

## Referee Comment (RC1) · Anonymous Referee #3 · 18 Aug 2017

The authors describe a new filter photometer which has been extensively laboratory and field tested. The manuscript is very concise and compact and should be published as soon as possible subject to the recommendations below.

It would be beneficial to the article to present more comparisons to a more varying set of different instruments for the measurement of Black Carbon or the determination of the aerosol absorption coefficient. The comparison to the PSAP is important for continuity, but intercomparisons of the CLAP with other instruments have been performed and could be reported here.

The fixed parameters in Eqs. 3 and 4 are a "widely accepted correction scheme" for the PSAP. This does not necessarily mean the most accurate one for each particular site, as the filter-particle interaction in filter photometers depends on the sample, not just the

filter properties. A wider and more comprehensive discussion on the applicability of a fixed scheme for the sites where the PSAPs and CLAPs were installed would benefit the readers, especially the discussion of the limitations of a fixed scheme. This would be appropriate for the article and should be included. Please see also the comment below (page 1, line 32).

Detailed comments

Page 1, Line 7: The authors mention the "measurement" of the absorption coefficient. In fact the measurement is of attenuation and the absorption coefficient is "determined", using a correction scheme. I would recommend to use "determination" here and throughout the manuscript.

P 1, L 16: ". . . light absorption measurements are recommended for all stations in the [GAW] network. . .". Please add a reference.

P 1, L 32: ". . .the Aethalometer doesn't yet have a widely accepted correction scheme." Several sentences describing the different schemes would help, including the most widely used ones for the PSAP (Bond et al., 1999; also in light of Virkkula, 2010) and the Aethalometer (Weingartner et al., 2003). The discussion on the separation of the influences of the loading effects and the multi-scattering on the measurement would be highly appropriate, especially since the final goal is to determine the absorption coefficient.

P 2, L 8: ". . . 10 mm aerodynamic diameter;" 10 micro-meters.

P 2, L 11: Filter loading and multiple scattering are not "errors", but effects that need to be taken into account when the absorption coefficient is determined.

P 3, L 21: The influence of the internal heater on the sample should be elaborated on, for example, temperature specified and the influence on the volatile fraction of the sample discussed.

P 4, L 1: Specify the model of the TSI nephelometer.

P 4, L 13: Since there is no flow control, the needle valve will have a varying influence on the flow which will depend on the loading of the sample spot. What is the variation of the flow at the start and end of the sampling through one of the 8 spots? Does the variation of the face velocity matter for the correction scheme?

P 4, L 24: Please add the model of the external pump which was used in the laboratory and ambient experiments.

P 5, L 32-34: Please explicitly mention that the sample interval delta-t is the 8 hour period.

P 6, L27-36: For the readers not familiar with the Springston and Sedlacek (2007) paper, please add a description of Cases I-III and the reason for the different slopes.

P 7, L 17: "Fig. 8 shows the results of evaluating Eq. 6..." It is Eq. 7 that is evaluated.

P 8, L 12: The intercept in the regression here is not fixed, but it is in a later comparison (PSAP to CLAP). It would be more concise to use a single approach, given the small intercept, I recommend forcing the regression through the origin in all regressions in the manuscript.

P 10, L 2: "... has proven to be...". Please add references, especially for the aircraft operation.

P 10, L. 13: "...yielding results that are about 25% higher..." Please specify the parameter, rather than "results".

Figure 10: "The slope of the regression line is 1.04 ..." Is this the inverse of the regression line slope mentioned in the text (are the x and y axes different than the ones referred to in the text)?

I applaud the authors in releasing the technical information of the new instrument, including the technical drawings, PCB layouts and the source code. It would be clearer if the license for this documentation would be explicitly mentioned.

**References**

Bond, T. C., Anderson, T. L., and Campbell, D.: Calibration and intercomparison of filter-based measurements of visible light absorption by aerosols, Aerosol Sci. Tech. 30(6): 582-600, 1999.

Virkkula, A.: Correction of the Calibration of the 3-wavelength Particle Soot Absorption Photometer ($3\lambda$ PSAP), Aerosol Science and Technology, 44: 8, 706-712, 2010.

Weingartner, E., Saathoff, H., Schnaiter, M., Streit, N., Bitnar, B., and Baltensperger, U.: Absorption of light by soot particles: determination of the absorption coefficient by means of aethalometers, J. Aerosol Sci., 34, 1445–1463, doi:10.1016/S0021-8502(03)00359-8, 2003.

---

## Referee Comment (RC2) · Anonymous Referee #2 · 24 Aug 2017

General

The paper describes the technical details and performance of the CLAP, an instrument that is developed at NOAA for measuring light absorption by particles. The instrument is basically a new version of the already almost 20-year-old PSAP. It is excellent that the authors have continued the development since the original PSAP was probably the best instrument in the market for long-term absorption measurements in background conditions but it has the problem of manually changing filter spot after every time the transmittance decreased below a certain limit. And the manufacturer does not make them any more. The paper shows that the instrument obviously works very well. The noise of the instrument is low, the noise characteristics have been described in detail. There is also an intercomparison of PSAPs and CLAPS at 17 stations which is an

impressive number, not possible to accomplish by many if any other group. The comparison shows that the two instruments compare well which is important for switching from one to the other at long-term monitoring sites. The paper is very well written and I can recommend publishing it at AMT. I only have a few comments or questions.

Detailed comments

1) P3,L27-28. About the flow through two reference spots. "the reference measurement alternates between the two...". Is this done with a magnetic valve that switches continuously or what? How frequently? Doesn't this create a pulsating flow? I don't quite understand.

2) It is mentioned in many pages that there is a heater but no details are given. In section 2 there should be a description of it. How is this done, how warm, where, and how stable is it? Potentially this is important and could be included in all filter-based instruments.

2) Section 3.3, Spot area. I like this method for defining the area. Traditionally people measure it manually for all filter-based instruments, this is more scientific. I suppose you have made some comparison with the manual way – please report the results of that also.

---

## Referee Comment (RC3) · Anonymous Referee #1 · 31 Aug 2017

GENERAL COMMENT

The manuscript describes a new absorption photometer, designed for the continuous measurement of the aerosol light absorption coefficient. The instrument is based on the well-established PSAP method and extends its approach for continuous measurement. The presented instrument fills an important gap in the available suit of instruments available for measuring aerosol light absorption since the PSAP will no longer be available. The manuscript is in the core focus of Atmos. Meas. Tech., it is well written, and is acceptable for publication after minor changes have been considered.

The main comment refers to the correction scheme applied to the CLAP attenuation coefficient data for converting it into absorption coefficients. In the abstract, the authors state that the improved performance of CLAP compared to PSAP is achieved by

means of an improved correction scheme. When reading the manuscript, the differences between the CLAP correction scheme and the Ogren (2010) scheme for PSAP is not clear. On the other hand, the authors provide a detailed analysis of instrument uncertainty terms which is clear and well presented, and results in an improved understanding of the uncertainty of absorption coefficients provided by CLAP, compared to those provided by PSAP. Furthermore, the determination of the spot area is significantly improved, which also helps improving the data quality. Overall, a summarizing section would be helpful, which highlights the new features of the full correction scheme.

The comparison between CLAP and PSAP (Fig. 10) is convincing. Looking at Fig. 11 there seems to be no systematic behavior of the scatter of regression slopes between CLAP and PSAP. To fully understand the meaning of this observation it would be good to see plots similar to Fig. 10 for a couple of stations with slopes close to unity and at the extreme values. Whereas the average ratio of PSAP to CLAP absorption coefficients is 0.94 and the scatter of ratios does not show any dependence on the attenuation coefficient, it would be of interest if the scatter of station correlations can be related to intensive aerosol properties like average single-scattering albedo etc. The overall question here is whether the correlation between CLAP and PSAP is robust for each station, but with a slope different from unity. If this is the case, then a discussion of this finding would be helpful.

MINOR COMMENTS

Page 1, line 32. An overview of existing correction schemes for the Aethalometer are compiled in Collaud Coen et al. (2010). This paper should be cited.

Page 2, line 8: It should read "10 $\mu$m" instead of "10 mm".

Page 2, line 10: The reference to Ogren's comment (Ogren, 2010) on the calibration of filter-based instruments might be given here since it describes the method used for correcting PSAP data at multiple wavelengths.

Page 2, line 11: Filter-loading and multiple-scattering effects should not be referred to as errors of filter-based methods but as intrinsic effects given by the physical processes involved in the signal generation.

Page 3, line 15: The correction (Virkkula, 2010) to Virkkula's PSAP algorithms (Virkkula et al., 2005) should also be referenced here.

Page 6, line 27-36: This section on CLAP noise versus PSAP noise needs to be further elaborated. So far, only experts knowing the work of SS07 can fully understand this section. The overall conclusion of this section should be mentioned more explicitly.

Page 7, line 7: The noise measurements are presented in Section 3.6.

Page 22,Fig. 10: Why not adding detailed values incl. uncertainties for slope and offset of the regression line? Is the offset statistically significant?

REFERENCES

Collaud Coen, M., Weingartner, E., Apituley, A., Ceburnis, D., Fierz-Schmidhauser, R., Flentje, H., Henzing, J. S., Jennings, S. G., Moerman, M., Petzold, A., Schmid, O., and Baltensperger, U.: Minimizing light absorption measurement artifacts of the Aethalometer: evaluation of five correction algorithms, Atmos. Meas. Tech., 3, 457-474, doi: 10.5194/amt-3-457-2010, 2010.

Ogren, J. A.: Comment on "Calibration and Intercomparison of Filter-Based Measurements of Visible Light Absorption by Aerosols", Aerosol Science and Technology, 44, 589 - 591, 2010.

Virkkula, A., Ahlquist, N. C., Covert, D. S., Arnott, W. P., Sheridan, P. J., Quinn, P. K., and Coffman, D. J.: Modification, calibration and a field test of an instrument for measuring light absorption by particles, Aerosol Sci. Technol., 39, 68-83, 2005.

Virkkula, A.: Correction of the Calibration of the 3-wavelength Particle Soot Absorption Photometer (3-wavelength PSAP), Aerosol Sci. Technol., 44, 706-712, doi:

10.1080/02786826.2010.482110, 2010.

---

## Author Comment (AC1) · 4 Oct 2017

***Reviewers' comments and authors' replies on "Continuous Light Absorption Photometer for Long-Term Studies" by John A. Ogren et al.***

We thank the reviewers for the time they spent carefully reading the manuscript and constructively commenting on the paper.

In the text below, reviewers' comments are shown in italicized sans-serif font and authors' responses are shown in serif font.

**Anonymous Referee #1**

*GENERAL COMMENT*

*The manuscript describes a new absorption photometer, designed for the continuous measurement of the aerosol light absorption coefficient. The instrument is based on the well-established PSAP method and extends its approach for continuous measurement. The presented instrument fills an important gap in the available suit of instruments available for measuring aerosol light absorption since the PSAP will no longer be available. The manuscript is in the core focus of Atmos. Meas. Tech., it is well written, and is acceptable for publication after minor changes have been considered.*

*The main comment refers to the correction scheme applied to the CLAP attenuation coefficient data for converting it into absorption coefficients. In the abstract, the authors state that the improved performance of CLAP compared to PSAP is achieved by means of an improved correction scheme. When reading the manuscript, the differences between the CLAP correction scheme and the Ogren (2010) scheme for PSAP is not clear.*

The reason why "the differences … is not clear" is because there are no differences; the same correction scheme was used for both PSAP and CLAP. The Reviewer incorrectly interpreted the abstract text.

*On the other hand, the authors provide a detailed analysis of instrument uncertainty terms which is clear and well presented, and results in an improved understanding of the uncertainty of absorption coefficients provided by CLAP, compared to those provided by PSAP. Furthermore, the determination of the spot area is significantly improved, which also helps improving the data quality. Overall, a summarizing section would be helpful, which highlights the new features of the full correction scheme.*

Such a summarizing section is not needed, because no new features of the Bond et al. (1999) and Ogren (2010) correction schemes are introduced in the paper.

*The comparison between CLAP and PSAP (Fig. 10) is convincing. Looking at Fig. 11 there seems to be no systematic behavior of the scatter of regression slopes between CLAP and PSAP. To fully understand the meaning*

*of this observation it would be good to see plots similar to Fig. 10 for a couple of stations with slopes close to unity and at the extreme values.*

Figure 10 has been expanded to include four stations that span nearly the full range of slopes. The analysis was repeated with different filtering of outliers, which has yielded slightly different results (but not meaningfully different, considering the uncertainty of the data).

*Whereas the average ratio of PSAP to CLAP absorption coefficients is 0.94 and the scatter of ratios does not show any dependence on the attenuation coefficient, it would be of interest if the scatter of station correlations can be related to intensive aerosol properties like average single-scattering albedo etc. The overall question here is whether the correlation between CLAP and PSAP is robust for each station, but with a slope different from unity. If this is the case, then a discussion of this finding would be helpful.*

It is not clear what the Reviewer means by "robust", but from the rest of this comment is would appear he/she means that the slope of the regression line is related to properties of the aerosol at the station. We looked at plots of the slope of the regression line versus various intensive properties (single-scattering albedo, backscatter fraction, scattering and absorption Ångström exponents), and did not see any clear relationships.

The main conclusion of the analysis of the regression slopes is given in the revised last sentence of section 3.7: "The average ratio of 0.91 derived here is thus indistinguishable from unity with better than 95% confidence, indicating that measurements of the attenuation coefficient with the CLAP and PSAP are quantitatively consistent within the uncertainty of the measurements."

*MINOR COMMENTS*

*Page 1, line 32. An overview of existing correction schemes for the Aethalometer are compiled in Collaud Coen et al. (2010). This paper should be cited.*

The citation has been added.

*Page 2, line 8: It should read "10 µm" instead of "10 mm".*

The error has been corrected.

*Page 2, line 10: The reference to Ogren's comment (Ogren, 2010) on the calibration of filter-based instruments might be given here since it describes the method used for correcting PSAP data at multiple wavelengths.*

The citation on p.2 is referring to the instrument, not the correction scheme. The Ogren (2010) citation is correctly located in the text where the correction scheme is discussed.

*Page 2, line 11: Filter-loading and multiple-scattering effects should not be referred to as errors of filter-based methods but as intrinsic effects given by the physical processes involved in the signal generation.*

The wording has been changed from "errors" to "interfering effects"

*Page 3, line 15: The correction (Virkkula, 2010) to Virkkula's PSAP algorithms (Virkkula et al., 2005) should also be referenced here.*

The citation has been added.

*Page 6, line 27-36: This section on CLAP noise versus PSAP noise needs to be further elaborated. So far, only experts knowing the work of SS07 can fully understand this section. The overall conclusion of this section should be mentioned more explicitly.*

Further elaboration of the approach and results of SS07 is beyond the scope of this paper. The method for calculating the CLAP noise is comparable to the Case II method in SS07, and the SS07 Case II PSAP results are the only ones that are compared with the CLAP noise data. Expert knowledge of SS07 is not required to understand these comparisons of PSAP noise with CLAP noise, as only one averaging technique is discussed for the CLAP data. An explicit conclusion of the CLAP-PSAP noise comparisons has been added: "As a result, about all that can be concluded is that CLAP and PSAP noise levels are similar, to within a factor of two."

*Page 7, line 7: The noise measurements are presented in Section 3.6.*

The error has been corrected.

*Page 22, Fig. 10: Why not adding detailed values incl. uncertainties for slope and offset of the regression line? Is the offset statistically significant?*

The figure has been expanded to include the slopes and intercepts of the regression lines, along with their uncertainties. The intercepts are statistically significantly different from zero, but the values are generally below the noise level of hourly averages of attenuation coefficients from the CLAP and PSAP. The text has been modified accordingly.

*REFERENCES*

*Collaud Coen, M., Weingartner, E., Apituley, A., Ceburnis, D., Fierz-Schmidhauser, R., Flentje, H., Henzing, J. S., Jennings, S. G., Moerman, M., Petzold, A., Schmid, O., and Baltensperger, U.: Minimizing light absorption measurement artifacts of the Aethalometer: evaluation of five correction algorithms, Atmos. Meas. Tech., 3, 457- 474, doi: 10.5194/amt-3-457-2010, 2010.*

*Ogren, J. A.: Comment on "Calibration and Intercomparison of Filter-Based Measurements of Visible Light Absorption by Aerosols", Aerosol Science and Technology, 44, 589 - 591, 2010.*

*Virkkula, A., Ahlquist, N. C., Covert, D. S., Arnott, W. P., Sheridan, P. J., Quinn, P. K., and Coffman, D. J.: Modification, calibration and a field test of an instrument for measuring light absorption by particles, Aerosol Sci. Technol., 39, 68-83, 2005.*

*Virkkula, A.: Correction of the Calibration of the 3-wavelength Particle Soot Absorption Photometer (3-wavelength PSAP), Aerosol Sci. Technol., 44, 706-712, doi: 10.1080/02786826.2010.482110, 2010.*

**Anonymous Referee #2**

*General*

*The paper describes the technical details and performance of the CLAP, an instrument that is developed at NOAA for measuring light absorption by particles. The instrument is basically a new version of the already almost 20-year-old PSAP. It is excellent that the authors have continued the development since the original PSAP was probably the best instrument in the market for long-term absorption measurements in background conditions but it has the problem of manually changing filter spot after every time the transmittance decreased below a certain limit. And the manufacturer does not make them any more. The paper shows that the instrument obviously works very well. The noise of the instrument is low, the noise characteristics have been described in detail. There is also an intercomparison of PSAPs and CLAPS at 17 stations which is an impressive number, not possible to accomplish by many if any other group. The comparison shows that the two instruments compare well which is important for switching from one to the other at long-term monitoring sites. The paper is very well written and I can recommend publishing it at AMT. I only have a few comments or questions.*

*Detailed comments*

*P3,L27-28. About the flow through two reference spots. "the reference measurement alternates between the two...". Is this done with a magnetic valve that switches continuously or what? How frequently? Doesn't this create a pulsating flow? I don't quite understand.*

The text has been changed to clarify that the reference spot is changed to the alternate spot each time the sample spot is changed. This is an infrequent event and there are no problems with pulsating flow.

*It is mentioned in many pages that there is a heater but no details are given. In section 2 there should be a description of it. How is this done, how warm, where, and how stable is it? Potentially this is important and could be included in all filter-based instruments.*

A description of the heater and typical operating temperatures has been added: "The internal heater consists of two 78 ohm thin-film heaters glued to the bottom of the aluminum plate that holds the filter (Fig 1). The temperature of this plate is measured with a thermistor and the microprocessor controls the power supplied to the heater in order to maintain the desired temperature. The plate temperature is normally set to 35-39 ºC, which is sufficient to reduce the relative humidity to below 40% most of the time. The temperature is typically

maintained within 0.1 °C of the set point temperature, except in cases where the heat dissipated by the solenoid valves and electronics inside the CLAP is sufficient to cause the plate temperature to exceed the set point temperature; such cases occur in warm laboratories, and require an operating temperature of 39°C to achieve stable operation. Some losses of semi-volatile species might occur at these temperatures. The effects of such possible losses on the measured attenuation coefficient are unknown but assumed to be small because the dominant light-absorbing species (black carbon) is not volatile and the amount of heating is relatively small; however, this is an area where further research would be useful."

*2) Section 3.3, Spot area. I like this method for defining the area. Traditionally people measure it manually for all filter-based instruments, this is more scientific. I suppose you have made some comparison with the manual way – please report the results of that also.*

Good point. We had not previously done a rigorous comparison of the manual determination of spot diameter with a reticle versus the automatic determination with photographic image analysis. We manually measured the diameter of 12 randomly-chosen, archived CLAP spots (five independent analysts, five measurements per spot) with an optical reticle and compared the results with the automated analyses. Overall, the spot area determined manually was 1.3% smaller than the automated analysis, which is within the uncertainty of the automated analysis. A summary of this comparison is now included in the text.

**Anonymous Referee #3**

*The authors describe a new filter photometer which has been extensively laboratory and field tested. The manuscript is very concise and compact and should be published as soon as possible subject to the recommendations below.*

*It would be beneficial to the article to present more comparisons to a more varying set of different instruments for the measurement of Black Carbon or the determination of the aerosol absorption coefficient. The comparison to the PSAP is important for continuity, but intercomparisons of the CLAP with other instruments have been performed and could be reported here.*

The primary focus of the current paper is to describe the CLAP and establish its continuity with the instrument it replaces (PSAP). Comparisons with other instruments are beyond the scope of this paper. We are working on analysis of an experiment conducted in Leipzig in 2016 where a variety of instruments, including CLAP, PSAP, MAAP, and AE33 Aethalometer, were operated in parallel with reference instruments for measuring absorption coefficient. Results from that experiment will be the subject of future papers.

*The fixed parameters in Eqs. 3 and 4 are a "widely accepted correction scheme" for the PSAP. This does not necessarily mean the most accurate*

*one for each particular site, as the filter-particle interaction in filter photometers depends on the sample, not just the filter properties. A wider and more comprehensive discussion on the applicability of a fixed scheme for the sites where the PSAPs and CLAPs were installed would benefit the readers, especially the discussion of the limitations of a fixed scheme. This would be appropriate for the article and should be included. Please see also the comment below (page 1, line 32).*

We wish that we had the data to discuss the limitations of a fixed correction scheme. An analysis of those limitations would require data from a reference instrument (e.g., photoacoustic) for comparison with the corrected CLAP data, which we do not have at our field stations. Even if we had the data, the requested analysis and discussion is beyond the scope of the present paper.

*Detailed comments*

*Page 1, Line 7: The authors mention the "measurement" of the absorption coefficient. In fact the measurement is of attenuation and the absorption coefficient is "determined", using a correction scheme. I would recommend to use "determination" here and throughout the manuscript.*

We agree that the filter-based methods measure attenuation and use a correction scheme to determine the absorption coefficient. We have changed the wording as recommended throughout the text, sometimes using "observation" rather than "determination".

*P 1, L 16: ". . . light absorption measurements are recommended for all stations in the [GAW] network. . .". Please add a reference.*

This citation has been added.

*P 1, L 32: ". . .the Aethalometer doesn't yet have a widely accepted correction scheme." Several sentences describing the different schemes would help, including the most widely used ones for the PSAP (Bond et al., 1999; also in light of Virkkula, 2010) and the Aethalometer (Weingartner et al., 2003). The discussion on the separation of the influences of the loading effects and the multi-scattering on the measurement would be highly appropriate, especially since the final goal is to determine the absorption coefficient.*

The primary goals of the paper are to present the design and performance of the CLAP, and to compare the CLAP measurements with the PSAP. Discussion and evaluation of the correction schemes used to derive absorption coefficient from attenuation coefficient are outside the scope of the paper.

*P 2, L 8: ". . . 10 mm aerodynamic diameter;" 10 micro-meters.*

The error has been corrected.

*P 2, L 11: Filter loading and multiple scattering are not "errors", but effects that need to be taken into account when the absorption coefficient is determined.*

The wording has been changed from "errors" to "interfering effects"

*P 3, L 21: The influence of the internal heater on the sample should be elaborated on, for example, temperature specified and the influence on the volatile fraction of the sample discussed.*

A description of the heater and typical operating temperatures has been added: "The internal heater consists of two 78 ohm thin-film heaters glued to the bottom of the aluminum plate that holds the filter (Fig 1). The temperature of this plate is measured with a thermistor and the microprocessor controls the power supplied to the heater in order to maintain the desired temperature. The plate temperature is normally set to 35-39 ºC, which is sufficient to reduce the relative humidity to below 40% most of the time. The temperature is typically maintained within 0.1 ºC of the set point temperature, except in cases where the heat dissipated by the solenoid valves and electronics inside the CLAP is sufficient to cause the plate temperature to exceed the set point temperature; such cases occur in warm laboratories, and require an operating temperature of 39ºC to achieve stable operation. Some losses of semi-volatile species might occur at these temperatures. The effects of such possible losses on the measured attenuation coefficient are unknown but assumed to be small because the dominant light-absorbing species (black carbon) is not volatile and the amount of heating is relatively small; however, this is an area where further research would be useful."

*P 4, L 1: Specify the model of the TSI nephelometer.*

The model number has been added.

*P 4, L 13: Since there is no flow control, the needle valve will have a varying influence on the flow which will depend on the loading of the sample spot. What is the variation of the flow at the start and end of the sampling through one of the 8 spots? Does the variation of the face velocity matter for the correction scheme?*

Routine quality-control review of the data has not revealed problems with flow stability or drift. One month of data from eight instruments (312 spots) yielded a median coefficient of variation of the flow rate of 0.4%. The text has been expanded to state that "active flow control is not needed, as the flow rate typically varies by less than 1% during sampling".

*P 4, L 24: Please add the model of the external pump which was used in the laboratory and ambient experiments.*

A variety of pumps have been used. The text has been expanded to indicate that carbon vane pumps are commonly used with the CLAP.

*P 5, L 32-34: Please explicitly mention that the sample interval delta-t is the 8 hour period.*

The wording has been changed to clarify that the value of Δt ranged from 1s to about 3 h.  The 8 hour period is the time that each spot was sampled, and is not related to the value of Δt (other than it establishes the upper limit for Δt).

*P 6, L27-36: For the readers not familiar with the Springston and Sedlacek (2007) paper, please add a description of Cases I-III and the reason for the different slopes.*

Further elaboration of the approach and results of SS07 is beyond the scope of this paper, and as already mentioned in the text, many of the results of SS07 are specific to the peculiarities of the internal data processing and serial output of the PSAP. The key information from SS07 that is relevant to the CLAP-PSAP noise comparison is already presented, but the text has been expanded to make it clear that the single PSAP tested by SS07 had a lower noise level than the six PSAPs tested by Müller et al. (2011) and than the CLAP.

*P 7, L 17: "Fig. 8 shows the results of evaluating Eq. 6. . ." It is Eq. 7 that is evaluated.*

The error has been corrected.

*P 8, L 12: The intercept in the regression here is not fixed, but it is in a later comparison (PSAP to CLAP). It would be more concise to use a single approach, given the small intercept, I recommend forcing the regression through the origin in all regressions in the manuscript.*

We repeated the regression analysis using ordinary least-squares with the regression line forced through the origin, and found a mean slope of 0.90 (vs. 0.91 with the orthogonal regression). The discussion has been expanded to include this result, and to give our reasons for using the slope of the orthogonal regression analysis. However, considering the combined uncertainties of the CLAP and PSAP, both regression analysis techniques yield slopes that are not statistically different from unity.

*P 10, L 2: ". . . has proven to be. . .". Please add references, especially for the aircraft operation.*

The CLAP has been used in several aircraft campaigns but the results have not yet been published. References to ground-based studies using the CLAP have been added.

*P 10, L. 13: ". . .yielding results that are about 25% higher. . ." Please specify the parameter, rather than "results".*

The wording has been changed to indicate that the parameter is attenuation coefficient.

*Figure 10: "The slope of the regression line is 1.04 . . ." Is this the inverse of the regression line slope mentioned in the text (are the x and y axes different than the ones referred to in the text)?*

The value of 1.04 is correct. Figure 10 has been revised to include results from four stations that span the full range of slopes, based on the comment from Reviewer #1. The case shown in the original Figure 10 is not included in the new Figure 10 because the slope was not sufficiently close to the target slopes for the four panes in the new Figure 10 (1.08, 0.96, 0.84, 0.72).

*I applaud the authors in releasing the technical information of the new instrument, including the technical drawings, PCB layouts and the source code. It would be clearer if the license for this documentation would be explicitly mentioned.*

The applicable sentence as been appended with "under the terms of the GNU General Public License v2".

*References*

*Bond, T. C., Anderson, T. L., and Campbell, D.: Calibration and intercomparison of filter-based measurements of visible light absorption by aerosols, Aerosol Sci. Tech. 30(6): 582-600, 1999.*

*Virkkula, A.: Correction of the Calibration of the 3-wavelength Particle Soot Absorption Photometer (3λ PSAP), Aerosol Science and Technology, 44: 8, 706-712, 2010.*

*Weingartner, E., Saathoff, H., Schnaiter, M., Streit, N., Bitnar, B., and Baltensperger, U.: Absorption of light by soot particles: determination of the absorption coefficient by means of aethalometers, J. Aerosol Sci., 34, 1445–1463, doi:10.1016/S0021- 8502(03)00359-8, 2003.*

---

## Author Response (AR2)

Associate Editor Decision: Publish subject to minor revisions (review by editor) (21 Oct 2017) by Willy Maenhaut

Comments to the Author:

The authors have reasonably addressed the comments of the three anonymous referees and they have modified their manuscript accordingly. However, alterations are needed in the main text before the manuscript can be published in AMT.

Response to the Editor:

Thank you for accepting our responses to the reviewers. All the alterations recommended on 21 Oct 2017 were made as suggested, as indicated by a "√" symbol at the start of each line below. The only place where any elaboration is needed concerns the comment on Page 3, line 14, highlighted below.

√Page 1, line 7: Replace "of aerosol" by "of the aerosol".

√Page 1, line 27: Replace "measuring aerosol" by "measuring the aerosol".

√Page 1, line 30: Replace "of aerosol" by "of the aerosol".

√Page 1, line 31: Replace "at single" by "at a single".

15 √Page 1, line 32: Replace "doesn't yet" by "does not yet".

√Page 2, line 1: Insert a comma between "green" and "red".

√Page 2, lines 18, 20 and 21: Replace the period at the end of the line by a semicolon.

√Page 2, line 25: Replace "calculating light" by "calculating the light".

√Page 3, line 14: "in the uncertainties of ..." is unclear. Should it perhaps be "and the uncertainties of ..." instead? [JO: I added a

20 comma after $\sigma_a$ to clarify the meaning, which is that the uncertainties of K1 and K2 implicitly include the uncertainties of $f(\tau)$ and $\sigma_a$.]

√Page 5, line 6: Replace "sample efficiency" by "sampling efficiency".

√Page 5, line 15: Replace "Spot areas" by "The spot areas".

√Page 5, line 23: Replace "taken with" by "taken with a".

25 √Page 5, line 26: Replace "of spot" by "of the spot".

√Page 5, line 28: Replace "unaware of" by "unaware of the".

√Page 5, line 29: Replace "of spot" by "of the spot".

√Page 6, line 4: Replace "of attenuation" by "of the attenuation".

√Page 6, line 6: Replace "calculating precision" by "calculating the precision".

30 √Page 6, line 9: Replace "of repeated" by "of the repeated".

√Page 6, line 12: Replace "of CLAP" by "of the CLAP".

√Page 6, line 14: Replace "Noise" by "The noise" and replace "when" by "when the".

√Page 6, line 15: Replace "CLAP sample" by "the CLAP sample".

√Page 6, line 33: Replace "Input data" by "The input data".

35 √Page 6, line 38: Replace "Overall" by "The overall".

√Page 7, line 12: Replace "of PSAP" by "of the PSAP".

√Page 7, line 19: Replace "that CLAP" by "that the CLAP".

√Page 7, line 33: Replace "of attenuation" by "of the attenuation".

√Page 8, line 6: Replace "to uncertainties" by "to the uncertainties".

40 √Page 8, line 14: Replace "range of" by "range of the".

√Page 8, line 18: Replace "both PM10" by "both the PM10".

√Page 8, line 30: Replace "of CLAP" by "of the CLAP".

√Page 9, line 2: Replace "Results" by "The results".

√Page 9, line 15: Replace "of attenuation" by "of the attenuation".

5 √Page 10, line 3: Replace "of attenuation" by "of the attenuation".

√Page 10, line 14: Replace "of attenuation" by "of the attenuation".

√Page 10, line 23: Abbreviations and acronyms, here "EBC", should be defined (written full-out) when first used.

√Page 10, line 32: Replace "determining aerosol" by "determining the aerosol".

√Pages 12-14, References: There should be a comma after the abbreviated journal name and there should also be a comma (and

10 not a semicolon) after the journal volume.

√Page 13, line 36: Replace "S. R., and" by "S. R. and".

√Page 14, lines 1-2: "Virkkula, 2010" should come before "Virkkula et al., 2005".

√Page 15, heading of Table 1: Replace "of light" by "of the light".

√Page 15, heading of Table 2: Replace "of CLAP" by "of the CLAP".

15 √Page 16, caption of Figure 2: Replace "sample efficiency" by "sampling efficiency".

√Page 17, caption of Figure 3: Replace "of light source for blue, green, and red channels, and spectral sensitivity of detector" by

"of the light source for the blue, green, and red channels, and spectral sensitivity of the detectors".

√Page 19, caption of Figure 5, first line: Replace "of attenuation" by "of the attenuation" and replace "Line" by "The line".

√Page 19, caption of Figure 5, second line: Replace "of regression" by "of the regression".

20 √Page 20, caption of Figure 6, first line: Replace "of attenuation" by "of the attenuation" and replace "Thick" by "The thick".

√Page 20, caption of Figure 6, second line: Replace "thin" by "the thin".

√Page 20, caption of Figure 6, third line: Replace "red line represents" by "the red line represents the" and replace "blue" by "the

blue".

√Page 20, caption of Figure 6, fourth line: Replace "represents approximate" by "represents the approximate".

25 √Page 21, caption of Figure 7, first line: Replace "of attenuation" by "of the attenuation".

√Page 22, caption of Figure 8, first line: Replace "of absorption" by "of the absorption" and replace "of attenuation" by "of the

attenuation".

√Page 22, caption of Figure 8, second line: Replace "values of" by "values of the".

√Page 22, caption of Figure 8, third line: Place a period at the end of the line.

30 √Page 23, caption of Figure 9, first line: Replace "uncertainty of" by "uncertainty of the".

√Page 24, caption of Figure 10, sixth line: Replace "in gray" by "in gray and dashed".

√Page 25, caption of Figure 11, second line: Delete the "4" at the end of the line.

[revised manuscript text omitted]